# The Effect of Energy Use Rights Trading Policy on Environmental Performance: Evidence from Chinese 262 Cities

**DOI:** 10.3390/ijerph20043570

**Published:** 2023-02-17

**Authors:** Dong Le, Fei Ren, Yusong Li

**Affiliations:** School of Economics, Zhongnan University of Economics and Law, Wuhan 430073, China

**Keywords:** energy policy, environmental performance, double difference method, mediating effect, market development level, technological innovation

## Abstract

This study provides empirical evidence and policy inspiration for China to implement the energy use rights trading policy. Using 262 cities in China from 2005 to 2019 as samples, we employed the double difference method and mediation analysis to empirically measure the impact of energy use rights trading policy on environmental performance. First, energy use rights trading policy can improve urban environmental performance. This conclusion is valid as per the endogeneity test, parallel trend test, PSM-DID test, placebo test, and triple difference method. Second, heterogeneity analysis shows that the effect of the energy use rights trading policy on urban environmental performance will be different by the size of population. Energy use rights trading policy has the greatest effect on the environmental performance of resource-based cities. Meanwhile, compared to non-industrial base, the effect of the energy use rights trading policy on environmental performance is more pronounced in cities with older industrial base. Third, the mechanism test using the mediation effect model proved that the impact of energy use rights trading policy on environmental performance is achieved by improving the level of marketization and technological innovation.

## 1. Introduction

Although the economy of China has become the second largest in the world, and its per capita income ranks among middle-and high-income economies in the world after years of development, serious environmental pollution and ecological degradation caused by years of extensive and rapid growth cannot be ignored. According to the China Ecological and Environmental Bulletin (2021) [1], coal consumption rose 4.6 percent in 2021, accounting for 56 percent of total energy consumption, and 43.1 percent of China’s 339 cities still exceeded the air quality standard. The report of the 19th National Congress of the Communist Party of China also emphasized the control of the total amount of energy use and the improvement of energy utilization efficiency as one of the major problems to be overcome in the construction of socialist ecological civilization in China.

However, early Chinese policies aimed at coordinating energy and the environment were primarily focused on command-based environmental regulation. In recent years, environmental protection market of China has gradually explored the trading system and other important fundamental innovation practices, and all kinds of incentive environmental policies have gradually emerged in accordance with the new trend of market-oriented reform.

In 2016, the Chinese government put forward the “Energy Use Rights Trading Policy” in a pioneering way and carried out trials of paid use and trading of energy use rights in Zhejiang, Fujian, Henan, and Sichuan provinces [2]. As an innovative input permit trading policy, the study on its environmental effects is of great practical significance not only for sustainable environmental development and policy promotion in China, but also for the shortage of energy economics research. Based on this, this paper focuses on 262 prefecture-level cities, explores the impact of energy use rights trading policy on environmental performance from the city level, and tries to answer the following three questions: (1) Does the trading of energy use rights affect environmental performance? (2) How does energy use rights trading policy affect environmental performance? (3) Does energy use rights trading policy have heterogeneous impact on environmental performance?

There are two kinds of literature closely related to this study. The first group focuses on the policy effects of environmental trading systems [3]. The existing literature has made many achievements in environmental trading policies, such as emission trading and carbon emission trading, and has empirically analyzed the Porter effect [4] and environmental dividends [5] generated by them. Since energy rights trading in China is still in its initial stage, there are relatively few empirical studies on energy rights trading policies in China. Some scholars focus on energy trading policy and carbon dioxide emission trading system linkage. Haiying L and Yu (2019) [6] proposed that the policy combination of energy use right and carbon dioxide emission right can cause a significant increase in the potential energy-savings and green total factor productivity. The other scholars have studied the economic dividends produced by the energy uses rights trading policy. Energy consumption and use intensity are thought to be reduced by energy right trading mode [7]. The White Certificate system in the European and American markets is very similar to the transaction of energy rights and its research mainly focuses on environmental effects [8], energy efficiency [9], and cost [10]. It has been discovered that the energy trading policy benefits the environment, energy efficiency, and the reduction in transaction costs.

The second category of literature focuses on the relationship between economic variables and environmental performances. Research results in this field are endless, including administrative policies [11], economic performance [12], regional environment [13], population size [14], and technology efficiency [15]. With the ongoing improvement of environmental policies, the research literature has begun to focus on the impact of energy and environmental policies on environmental performance in recent years. Zhou Yuan and Zhang Xiaodong Zhang (2018) [16] studied 694 enterprises in Huzhou and found that in the three years after the green policy was promulgated, the waste water discharge intensity of related enterprises was significantly reduced. Peitian et al. (2020) [17] studied China’s provincial panel data from 2006 to 2017 and found that environmental regulation can affect environmental performance through green technology innovation.

We discovered a shortage of the existing literature by combing and review of the literature: (1) the existing literature more ignored can use trade policy to promote the key role of environmental performance improvement, few articles are directly involved in trading on the influence of environmental performance, and lack of energy-using trading policy impact on the environmental performance of the mechanism analysis. (2) In terms of empirical research, the existing literature primarily focuses on the data from provincial or listed companies and lacks the quantitative measurement of the environmental performance of prefecture-level cities, which cannot provide empirical support for improving environmental performance through energy-use trading policies. Based on this information, this study examines the impact and transmission mechanism of energy rights trading policies on environmental performance using 262 urban data samples from 2005 to 2019, while also expanding the research on the heterogeneous impact of energy rights trading policies in different types of cities.

Compared with the general literature on energy use rights trading policy and environmental performance, the main contributions of this paper are threefold: (1) It is the first time to evaluate the policy effect of the energy trading pilot from the comprehensive perspective of environmental performance level and refine the research scale to the city level. (2) Rooted in the insufficient marketization and innovation levels that exist in the implementation of energy rights trading policies in China, the level of marketization development and technological innovation are included in the mediating effect model as mediating variables to make up for the lack of empirical experience in the environmental impact mechanism of the energy rights trading literature. (3) The city types are divided into cities with different levels of urban development, resource-based cities, and old industrial bases, and the heterogeneous impact of energy use rights trading on different types of cities is deeply analyzed, and the further work focus of energy use rights trading policy is clarified.

## 2. Materials and Methods

Based on the Coase property right theorem [18], the energy rights trading policy defines the rights and interests of energy use and uses the market to achieve the Pareto optimal state of the energy allocation efficiency of the enterprise body, in order to achieve energy conservation and emission reduction at the lowest cost and improve environmental performance [19]. During the policy implementation process, enterprises will prioritize improving technological innovation and reducing energy intensity in order to gain advantages in energy use right trading and obtain maximum benefits at the lowest possible cost. However, pollution control expenditure and investment in new technology research and the development of enterprises will inevitably increase the financing demand, and the level of marketization development will directly affect the difficulty of enterprises to obtain financing. Therefore, this study selects marketization development level and technological innovation as mediating variables to explore the impact mechanism of energy and rights trading on environmental performance.

### 2.1. Mechanism Analysis

Energy trading policies can improve the level of marketization and affect urban environmental performance. Since energy use rights policy shares enter the market as a “commodity”, its operation process will be affected by the insufficiency of market mechanisms, such as speculation, the lag between supply and demand, and other problems, thus the level of marketization is closely related to the implementation effect of energy use rights trading. First, imperfect competition is prevalent in the market for energy use rights in practice. Some enterprises take advantage of purchasing and storing rights, obviously exceeding their quota to seek profits or for future use [20]. Whether the trading of energy use rights can be carried out in an open, fair, and reasonable trading market smoothly depends on the market level, the relationship between market and government, the development of the factor market, and so on. The total level of marketization reflects the degree to which factors flow from low-efficiency to high-efficiency sectors, and there is a positive relationship between the level of marketization and enterprise productivity. Second, in the practice of energy use rights trading, some local governments have “excessive intervention” in enterprises within the domain due to the consideration of economic development, which directly affects the enthusiasm of quota trading among enterprises. This indicates that the government is the key to effectively coordinating with the market to establish and perfect the market mechanism of energy use right trading. Only by relying on moderate government intervention and continuous marketization can we better promote the implementation of energy use right trading policy, which is required to improve environmental performance. Thus, this paper proposes Hypotheses 1:

**Hypothesis** **1.**
*Energy use rights trading policy can improve the level of marketization development and have a positive effect on urban environmental performance.*


Energy trading can stimulate the innovative vitality of enterprises. Combined with Porter’s hypothesis and the enterprise profit maximization theory, the energy trading policy has a promoting effect on enterprise technological innovation. The specific effects are mainly reflected in the following two aspects: First, innovation is the fundamental way for enterprises to pursue profit maximization. The trading use right policy of energy restrains the energy consumption of enterprises and stimulates enterprises to reduce production costs. Technological innovation has the potential to improve enterprise energy efficiency while also indirectly improving environmental performance. Second, the “Porter hypothesis” holds that moderate environmental regulation can bring compensation to enterprises for innovation activities and stimulate enterprises to carry out technological innovation activities [21]. When the company’s energy consumption falls below the quota, the rich energy rights can be sold for profit and compensation for innovation. Furthermore, the government can benefit from the sale of energy use rights and invest in related energy conservation and emission reduction projects, which not only lowers enterprises’ expectations of innovation risks, but also improves environmental performance. Ideally, the implementation of energy use rights trading policy can stimulate the vitality of technological innovation of enterprises. Because it can promote enterprises to improve production technology for the purpose of reducing energy consumption, eliminate backward production capacity, and indirectly promote the improvement of environmental performance level. According to the above analysis, this study proposes Hypothesis 2.

**Hypothesis** **2.**
*Energy use rights trading policy can improve the level of technological innovation and have a positive impact on environmental performance.*


The difference in energy-saving potential will lead to different impacts of energy use rights trading policies in different pilot areas. As a result of the disparities in urban resource allocation, the improvement effect of energy use rights trading policies on environmental performance may be heterogeneous.

First, the size of the city will affect the implementation effect of the energy use rights trading policy. Large cities can produce the aggregation effect of factors, which is mainly reflected in the spillover of labor, capital, knowledge, and the decline of marginal production costs. The high concentration degree of the city and the advantages of high production specialization, and strong scientific and technological innovation ability brought by the agglomeration of manufacturers form the scale economy effect makes it easier to attract the inflow of foreign capital and human capital for technological innovation and marketization development. At the same time, excessive agglomeration will also lead to excessive resource development, energy consumption, and affect the improvement of environmental performance. Therefore, in cities of different sizes, whether the energy use rights trading policy can have the effect of energy conservation and emission reduction needs to be further investigated.

Second, urban resource richness will affect the implementation effect of the energy use rights policy. According to the notice on the Issuance of the National Sustainable Development Plan for Resource-based Cities (2013–2020) issued by The State Council [22], resource-based cities can be divided into four categories: growth, maturity, decline, and regeneration, and there is a nonlinear relationship between resource abundance and ecological efficiency [23]. Cities with insufficient resource endowments (recession) have the greatest motivation for transformational development and technological innovation. Therefore, the energy use rights trading policy has high impact. While abundant resources (growth) cities may rely too heavily on the low-end of resource-intensive industries, this can stifle technological progress and, to some extent, improve environmental performance. Therefore, the energy use rights trading policy may have a greater effect on the improvement of environmental performance in cities with declining resources, although its effect is not significant in cities with growing resources.

Third, different industrial structures will also lead to the heterogeneous impact of energy use rights trading policies on environmental performance. The old industrial cities rely too much on resource-intensive industries and suffer from extensive development patterns. In this way, the long-term overexpansion of the industrial sector led to capital accumulation, but also led to high energy consumption, high pollution, and stagnant production technology. Cities that are not old industrial bases have a reasonable industrial structure layout, a low reliance on energy consumption for economic development, and a high level of environmental performance. Therefore, the energy use rights trading policy has more space to grow in the old industrial base cities. Based on the discussion above, we propose Hypotheses 3:

**Hypothesis** **3.**
*In cities with higher average urban economic income, resource decline, and higher industrial structure, energy use rights trading policy can perform a greater role.*


### 2.2. Methodology and Data

#### 2.2.1. Model Design

##### Double Difference Method

Under the premise that the treatment group and the control group meet the assumption of a parallel trend, the double difference method can observe the changes in the treatment group before and after the policy, and accurately measure the average treatment effect before and after the policy. In September 2016, the National Development and Reform Commission issued the Pilot Program of Paid Use and Transaction of Energy Rights, requiring Zhejiang, Fujian, Henan, and Sichuan to be pilot provinces of energy use rights trading policy, which provided a good “quasi-natural experiment” for this study. The double difference method is adopted to evaluate the impact of energy use rights trading policies on environmental performance. Its main advantages are that it cannot only effectively eliminate the interference of other factors on dependent variables, but also the estimation bias caused by missing variables can be fixed in the model [24]. As a result, this study built environmental performance at the city level. Taking Zhejiang, Fujian, Henan, and Sichuan provinces as the treatment group, the double difference method was used to explore the impact of energy trading policies on environmental performance. The benchmark model is as follows:(1)$$\ln ep_{it}=\beta_{0}+\beta_{1}treat_{i}\times period_{t}+\alpha_{1}X_{it}+\mu_{i}+v_{t}+\varepsilon_{it}$$
where i and t represent the province and year, respectively, $\ln ep_{it}$ is the environmental performance; treat is the dummy variable of the energy-use rights trading policy. If the province is subjected to the policy intervention of the energy-use rights trading policy, then treat=1, otherwise treat=0; period is a time dummy variable, if the year is after 2016, period=1, otherwise, period=0. $\beta_{0}$ is a constant term, $X_{\mathrm{it}}$ is a series of control variables affecting environmental performance; and $\mu_{i}$ is the fixed effect of the city, controlling for all factors at the city level that do not change over time. $v_{t}$ represents time-fixed effects, controlling for factors that do not change over time at the time level, and $\varepsilon_{it}$ is the residual term. In the above equation, $\beta_{1}$ is the estimated coefficient concerned by this study. If $\beta_{1}$ is positive and significant, it means that the energy use rights trading policy improves the environmental performance of the pilot provinces.

##### Mediation Effect Test

Based on the theoretical analysis, it can be concluded that the energy use rights trading policy can improve environmental performance by improving the level of marketization development and technological innovation. In this section, the mediation effect test procedure proposed by Wen Zhonglin (2014) [25] and the Bootstrap method was combined to test the mediation effect [26]. To thoroughly investigate the specific mechanism of energy use rights trading policy to improve environmental performance from two perspectives: marketization development level and technological innovation. The specific model is set as follows:(2)$$M_{\mathrm{it}}=\beta_{0}+\beta_{2}treat_{i}\times {\mathrm{period}}_{t}+\alpha_{s}X_{it}+\mu_{i}+v_{t}+\varepsilon_{it}$$
(3)$$\ln ep_{it}=\beta_{0}+\beta_{3}treat_{i}\times period_{t}+\beta_{4}M_{it}+\alpha_{s}X_{it}+u_{i}+v_{t}+\varepsilon_{it}$$

$M_{\mathrm{it}}$ is the intermediary variable, which is the level of marketization development and technological innovation, respectively. Other variables are consistent with those of model (1). $\beta_{2}$,$\beta_{3}$,$\beta_{4}$ is the estimated coefficient concerned by this study. If $\beta_{2}$,$\beta_{3}$, and $\beta_{4}$ are positive and significant, it means that the mediating effect of marketization and technological innovation is affected.

#### 2.2.2. Data and Variable Construction

##### Explained Variable

The explained variable urban environmental performance, $\ln ep_{it}$ mainly refers to the algorithm of Pingfang Zhu and Zhengyu Zhang (2011) [27], and builds a comprehensive index of environmental performance based on a variety of pollution emissions to measure urban environmental quality. First, the emission intensity of environmental pollution in each city is calculated:(4)$$I_{\mathrm{ijt}}=\frac{P_{ijt}}{Y_{it}}/\frac{1}{n}\sum_{1}^{n}\frac{P_{ijt}}{Y_{it}}$$
where $I_{\mathrm{ijt}}$ represents the intensity of j kinds of emissions in city i in period t, $P_{\mathrm{ijt}}$ represents the emission of j kinds of pollutants in city i in period t, and $Y_{\mathrm{it}}$ represents the total industrial output value of city i in period t. In addition, the intensity index of environmental pollution emissions is averaged.
(5)$$I_{\mathrm{it}}=1/m\sum_{j=1}^{m}I_{ijt}$$

In this study, the intensity of urban environmental pollution emissions is measured using three major environmental pollution emissions: industrial sulfur dioxide emissions, industrial dust emissions, and emissions. Finally, the comprehensive index of urban environmental performance is calculated through three kinds of pollution emissions.
(6)$${\mathrm{ep}}_{it}=1/I_{\mathrm{it}}$$

The higher the value of ${\mathrm{ep}}_{it}$, the higher the urban environmental performance. To alleviate heteroscedastic property, the logarithm of the urban environmental performance is expressed as *Inep_it_*.

##### Mediating Variable

The mediating variables are technological innovation and marketization level. They are calculated as follows: Referring to the measurement method of Fan Gang’s marketization index, the marketization development index is calculated by using the indicators of the relationship between the government and the market; the development of non-state-owned economy; the development of a product market; the development of a factor market; the development of intermediary organizations; and the law. Technological innovation is measured by the sum of the number of authorized patents and the number of public utility model patents [28].

##### Control Variables

Since the factors affecting environmental performance are very complex, control variables are used to control the possible effects of other factors to alleviate the endogeneity problem caused by omitted variable bias.

The financial development level, industrial structure, government intervention degree, and urbanization level were included in the model by referring to the existing literature of Dan and Shaolin (2020) [29] and Fei and Xu (2022) [30]. Financial development ${\mathrm{level}}_{\mathrm{it}}$ = (Balance of deposits and loans of financial institutions at $\mathrm{year}-{\mathrm{end}}_{\mathrm{it}}$/the total population at the end of the year). The industrial structure is measured by the proportion of the added value of the tertiary industry in the gross regional product. The degree of government intervention (lfb) is the proportion of local fiscal expenditure in GDP. Urbanization level (urb) is measured by the proportion of urban permanent resident population in the total regional permanent resident population.

##### Descriptive Statistics of Data

Based on data continuity and availability, this study selects data from 2005 to 2019 in 262 cities of China for empirical analysis. The data is mainly from the China statistical yearbook, China city statistical yearbook, China energy statistical yearbook, macroeconomic database, database of EPS, the state intellectual property office of the website, etc., with the missing value replaced with the mean. Table 1 shows the descriptive statistics of the main variables.

## 3. Results

### 3.1. Benchmark Regression

In order to verify the direct impact of energy use rights trading policy on environmental performance, this study uses the difference method to estimate the impact of energy use rights trading policy on environmental performance, and the regression results are shown in Table 2. In column (1) of Table 2, only fixed effects of time and city are controlled, and no other control variables are added. In column (2), control variables, such as urbanization, industrial structure, government intervention degree, and financial development level, are added to control the possible result bias, while column (3) shows the results of dynamic the effect analysis.

According to the regression results in Table 2, the regression coefficient of energy use rights trading policy is significantly positive at the level of 1%, whether control variables are added or not, indicating that the implementation of energy use rights trading policy has significantly improved the environmental performance of pilot cities. Column (2) shows that the estimated coefficient of energy use rights is 0.284, indicating that compared with non-pilot cities, the development of energy use rights trading policy has improved the environmental performance by 28.4%.

In the dynamic effect analysis, the improvement of environmental performance level shows obvious effects during the implementation of the energy use rights trading policy; the impact was strongest in 2016. In the following years, energy use rights trading policy still had a positive effect on the improvement of environmental performance, but the effect decreased year by year. The research results in Table 2 show that the energy trading policy can promote the improvement of environmental performance, but its effect is decreasing. As a result, it is critical to investigate the specific impact path of energy use rights trading on environmental performance.

### 3.2. Analysis of Heterogeneity

#### 3.2.1. Heterogeneity of Urban Development Level

In this study, the city size is classified based on the population at the end of the year using the Notice on the Criteria of City Size Classification issued by The State Council in 2014 and the size of 262 sample cities. Cities with fewer than one million people are classified as small- and medium-sized, cities with one to five million people are classified as large cities, and cities with more than five million people are classified as megacities. Table 3 shows the impact of energy use rights trading policies on environmental performance at different city sizes. It can be seen that the energy use rights trading policy significantly improves the environmental performance of large and super-large cities but has no significant impact on small- and medium-sized cities. This indicates that in large and super large cities energy use rights trading policy is compatible with economies of scale generated by agglomeration, and mature and developed factor markets, which can promote urban green transformation and promote the improvement of environmental performance. Small- and medium-sized cities lag behind large and super-large cities in terms of development, and their ability to allocate resources is poor. As a result, the implementation of an energy use rights trading policy raises the cost of capital and labor input factors, making the cost greater than the benefit, which limits enterprise transformation to some extent. As a result, the environmental performance is not significantly improved. It is worth noting that the impact of energy use rights trading policy in large cities is higher than that in super-large cities. The possible reason is that the increase in economies of scale to a certain extent will lead to the over-exploitation of resources and the rise of management costs, which will inhibit the improvement of environmental performance.

#### 3.2.2. Heterogeneity of Different Types of Resource-Based Cities

Regarding the distribution and use of energy, its energy efficiency will be affected by the dynamic change of resource endowment. Most resource-based cities are in a state of inefficient energy use, and there are significant differences among different resource-based cities. The List of China’s 262 resource-based Cities (2013–2020) released by The State Council identified 262 resource-based cities, county-level cities or districts, which are classified into growing, mature, declining, and regenerative resource-based cities according to the degree of resource abundance. Table 4 reports the heterogeneity effect of different types of resource-based cities in the sample. According to the empirical analysis results, energy use rights trading policies can promote the improvement of environmental performance in both resource-based cities and non-resource-based cities. Among them, the energy use rights trading policy has the most significant impact on the environmental performance of declining cities, followed by regenerative and mature cities, and the impact is not significant in growing cities.

At present, it is urgent to carry out the urban green transformation and get rid of the original heavy industry development mode based on energy consumption. The implementation of the energy use rights trading policy can improve the level of technological innovation, help the development of local market, aid the cities with declining resources to adjust the industrial structure, and accelerate the transformation and development of cities. Therefore, the energy use rights trading policy has the strongest marginal effect on environmental performance in declining cities. Resource-based cities have essentially eliminated resource dependence. This indicates that they need to further optimize the economic structure, improve the level of opening-up and scientific and technological innovation, and accelerate the development of emerging industries.

Energy use rights trading policy can stimulate urban activity by improving the level of market development, promoting the green transformation of resource-renewable cities, and improving their environmental performance. The main development goal is to pay attention to ecological environmental protection and improve resource utilization. The implementation of the energy use rights trading policy promotes the development of local technology innovation level, but its reliance on energy will not change in the short term. Therefore, the effect of the energy use rights trading policy on environmental performance is not as significant as that of declining and regenerative cities. Growing cities are the primary stages in the supply and reserve of energy resources in our country. When the resource exploitation is in the rising stage, the city will pay attention to the energy industry structure in the process of development and technological innovation. As the market is also steadily developing, the energy use rights trading policy implementation will not significant impact the city.

#### 3.2.3. Heterogeneity between Old Industrial Base Cities and Non-Old Industrial Base Cities

The National Plan for the Adjustment and Renovation of Old Industrial Bases (2013–2022) identified 120 old industrial base cities or provincial capital cities under their jurisdiction [31]. Among them, some of the old industrial bases are important national energy bases. Overall, the old industrial bases are characterized by high energy consumption and high pollution, which seriously restricts the improvement of environmental performance. In order to explore the differences in the impact that industrial base cities have the on environmental performance, this study has taken non-old industrial base cities located in energy use right trading provinces as the experimental group and non-old industrial base cities in other provinces as the control group to study the energy utilization efficiency of emission trading system for non-old industrial base cities.

The empirical regression results in Table 5 show that the energy use rights trading policy has a greater impact on the environmental performance of the old industrial base. The possible reasons are as follows: First, the old industrial base has a large room for improvement in environmental performance and is more sensitive to environmental policies aimed at energy consumption; Second, the old industrial base cities refer to the cities that carried out large-scale economic construction in the early days after the founding of New China, with heavy industry as the main industrial cluster. With the development of China’s economy and science and technology, their market development level and technological innovation need to be improved. The energy use rights trading policy has injected new momentum and vitality into their transformation, which has significantly improved the level of environmental performance. However, the industrial structure of non-old industrial base cities is relatively “light”. The average energy consumption and pollution level are lower than those of old industrial base cities, and cleaner production technology has developed well. Therefore, the improvement of their environmental performance level is relatively limited under the promotion of energy use rights trading policy.

#### 3.2.4. Test for Endogeneity

The Double difference method through comparing the experimental group and the control group can overcome the problem of endogenous adeptly, but the premise is that the trading pilot cities can be selected at random in all places, but this is not the case in reality. The pilot cities choice for the energy use rights trading policy may be influenced by other potential factors of interference, which may interfere with the result of the double difference method. While from the point of environmental performance development level and geographic features, energy use rights trading policy appears to be random, but this article raises concerns about the same period of other potential confounding factors affecting the choice of the pilot cities, such as over time the unpredictable factors may be associated with policy variables. Considering the endogenous problems that exist in the model, IV-2SLS is used for regression analysis in this section. Based on this understanding, this paper chooses the air circulation coefficient as the instrumental variable for whether to include the pilot city of energy rights trading, following the practice of Hering and Poncet (2014) [32]. The construction method of air flow coefficient is as follows: where $VC_{it}$, $WS_{it}$, and $BLH_{it}$ represent air flow coefficient, wind speed, and atmospheric boundary height, respectively.
(7)$$VC_{it}=WS_{it}\times BLH_{\mathrm{it}}$$

When the total pollutant emission is fixed, the smaller the air circulation coefficient is, the higher the pollutant monitoring concentration is, and the more likely it is to be selected as a pilot city for energy rights trading. On the other hand, the air flow coefficient is affected by both wind speed and the height of the atmospheric boundary layer, and both the wind speed and atmospheric boundary layer are determined by complex meteorological systems and geographical conditions, thus well satisfying the exogeneity assumption of effective instrumental variables [33]. In addition, the use of air circulation coefficient has a very advantageous point. That is, the use of air circulation coefficient variable can reasonably control the spatial spillover effect of energy rights trading policy, to accurately identify the impact of government environmental governance on energy rights trading policy.

The estimated results of instrumental variables are shown in Table 6. Iv is the air circulation coefficient of instrumental variables, which represents the natural logarithm of the annual average air coefficient of sample cities. In the first stage of regression, the coefficient of the interaction term iv period between instrumental variables and time variables is significant at the 1% level. The first stage F score is 17.51, which is greater than 10, thus significantly ruling out the “weak instrumental variable” problem. The second-stage regression results show that the impact of energy use rights trading policies on environmental performance is similar to the benchmark regression reported in Table 2 in both direction and significance, which further verifies the positive effect of energy use rights trading policies on environmental performance. However, in terms of regression coefficient, the estimated coefficient of energy use rights trading increases in absolute value when compared to the benchmark regression, indicating that the potential endogeneity problem tends to underestimate the positive effect of energy use rights trading on environmental performance.

### 3.3. Test for Robustness

#### 3.3.1. Parallel Trend Test

The important premise of using the difference method is that the experimental group and the control group meet the parallel trend assumption; From Figure 1,we can know that the environmental performance remains at a relatively stable trend before the implementation of the energy use rights trading policy pilot. Specifically, using 2016 as the benchmark year, OLS-DID regression was performed separately for the explained variables in the first and last three years of this benchmark, and the results were consistent with the main regression. In order to test this hypothesis, this paper draws on the practice of Beck et al., and uses the dynamic differential method to test the parallel trend. On the basis of formula (1), the following model is constructed:(8)$$\ln ep_{it}=\beta_{0}+\sum_{j=2005}^{j=2019}\beta_{j}D_{i.j}+\alpha_{1}X_{it}+u_{it}+v_{it}+\varepsilon_{it}$$
where, $D_{i.j}$ is a series of dummy variables representing the Jth year, in which the pilot of Province i began to be implemented. In addition, 2005 is taken as the base period, that is, the dummy variable with year = 2005 is not included in Equation (4). $\beta_{j}$ represents the impact of the pilot energy use rights trading policy on environmental performance before and after the implementation, which is also the core explanatory variable concerned by this model. When year <2016, $\beta_{j}$ is used to test whether the article meets the parallel trend test. If $\beta_{j}$ is not significant, the hypothesis of the parallel trend test is confirmed. Otherwise, it does not conform to the parallel trend hypothesis. When j ≥ 2016, $\beta_{j}$ is used to investigate the dynamic effect of the energy trading policy pilot.

#### 3.3.2. Propensity Matching Score Difference Method

The difference method can help to effectively identify and evaluate the policy effects of the energy use rights trading policy. However, this identification strategy faces the problem of sample selection bias in the process of use. To be specific, when selecting pilot units, the central government will comprehensively consider the economic development level, industrial structure, energy-saving potential, environmental capacity, and other factors of each region, and prefer to carry out pilot work in places with a certain foundation. In this way, sample selection bias will cause an endogeneity problem, which will lead to bias in the regression results. In order to solve the problem of sample bias, this paper used Propensity Score Matching and difference methods to re-estimate the impact of energy use rights trading policy on environmental regulation. Column (1) of Table 7 reports the results of using PSM-DID estimation.

The regression results show that the energy use rights trading policy is still significantly positive at the significance level of 1%, which further validates the conclusion that the energy rights trading policy can significantly improve environmental performance.

#### 3.3.3. Placebo Test

A placebo test was conducted in prefecture-level cities in pilot provinces with allocation to further test whether the results of this paper are driven by unobservable factors at the province-city-year level. Specifically, this paper selects 36 cities from prefecture-level cities as the treatment group, assuming that these 36 cities have implemented the energy use right trading policy, and the rest of the regions are the control group. Random sampling ensures that the independent variable treat* period constructed in this paper has no effect on environmental performance. In this paper, 500 benchmark regressions were performed, and the mean of the estimated coefficients for all treat* period was found to be almost zero. The distribution of the 500 estimated coefficients and their associated *p*-values are further plotted in this study, as shown in Figure 2. The distribution is concentrated in the 0 point annex, and the *p* value of the majority of estimates is greater than 0.1, indicating that the regression results in this paper are robust.

#### 3.3.4. Triple Difference Method

Even after excluding interference from the focus of energy use rights trading policy, there are still concerns about the impact of other policies that were not considered on the empirical results. For example, the emission rights trading policies that have been piloted in key cities, such as Beijing, Tianjin, Shanghai, and Chongqing, are likely to have a similar impact on local environmental performance. In order to further eliminate these possible disturbances, the triple difference method was used to overcome this problem according to the practice of Ziying (2017) [34]. The following triple difference model is established on the basis of the basic model. Treat is the dummy variable of the energy trading policy. If the province is subjected to the policy intervention of energy trading policy, then treat=1, otherwise treat=0. Period is the dummy variable of time. If the year is after 2016, period=1; otherwise, period=0. Res indicates that the old industrial base, resource-based city, provincial capital city, and the second largest city of the province are set as the new treatment variables; assign a value of 1 after 2016 and a value of 0 for all others. $period_{t}\times treat_{t}\times res_{t}$ is the core explanatory variable of the triple difference method and represents the regions belonging to the old industrial base, resource-based city, provincial capital city, and the second largest city of the province after the implementation of the pilot policy. The estimated coefficient is $\beta_{1}$, which is the core variable concerned by this model, and the meanings of the other variables are the same as in Equation (1).

With the help of triple difference method, the above-mentioned key cities are controlled, and some other policies that have not been taken into account are further excluded, to obtain the net impact of energy use right trading policy on environmental performance. The results of triple difference are shown in column (2) of Table 7.

The regression results once more prove the previous research conclusion that energy use right trading has a highly significant impact on environmental performance.
(9)$$\begin{array}{l} lnep_{it}=\beta_{0}+\beta_{1}period_{t}\times treat_{t}\times {\mathrm{res}}_{t}+\beta_{2}period_{t}\times treat_{t}+\beta_{3}period_{t}\times res_{t}\\ +\beta_{4}treat_{t}\times res_{t}+\alpha_{1}X_{it}+u_{it}+v_{it}+\varepsilon_{it} \end{array}$$

#### 3.3.5. Replace the Explained Variable

Measures of environmental performance are mainly three kinds: main pollution emissions intensity is divided by the gross value of industrial output, which is a measure of the total industrial output value of major pollution emissions. Based on the measurement of the three kinds of emissions, this section replaces the total industrial output value with GDP, expressing a measure of the impact of major pollution emissions on GDP. The empirical analysis results are shown in column (3) of Table 7. The estimated coefficient of energy use rights trading policy is significantly positive at 1% level, confirming the reliability of the basic results.

#### 3.3.6. Control for the Impact of Other Policies

Many economic policies cross over or appear in parallel for a specific economic goal during China’s economic reform process. For example, during the sample period of this article, the National Development and Reform Commission carried out low-carbon emission trading pilot projects in provinces and regions in 2010 and 2011, respectively. The carbon emission trading system overlaps with energy use rights trading in terms of trading scope and table, which may affect the benchmark regression results of this paper. Based on this, the paper eliminated the pilot samples of low-carbon provinces and carbon emission rights trading, and conducted a re-regression to avoid the interference of relevant policies. The regression results are shown in column (4) of Table 7. As shown in Table 7, after excluding the impact of other policies, the estimated coefficient of the energy use rights trading policy is still significantly negative at the significance level of 1%, which indicates that the impact of energy-use trading policy on environmental regulation is robust.

## 4. Mechanism Analysis Discussion

This paper uses mediation method in order to further verify the role of the energy use rights trading policy plays in influencing environmental performance. Table 8 lists the verification results of the impact mechanism. It has been discovered that, in terms of improving environmental performance, the improvement effect of energy use rights trading policy on environmental performance is significantly affected by market and technological innovation, and that improving market development and technological innovation levels can significantly promote environmental performance improvement.

Columns (1) and (4) of Table 8 are the baseline regression analyses. Column (2) and column (3) of Table 8 examine whether marketization is a bridge for the energy use rights trading policy to promote environmental performance. In column (2), the coefficient of the energy use rights trading policy and the coefficient of market-oriented development level are both significantly positive. In column (3), the energy use rights trading policy has a significant positive promoting effect on market-oriented water, indicating that the energy use rights trading policy can promote the improvement of environmental performance through the improvement of marketization level. Specifically, if all other factors remain constant, every unit increase in energy use rights trading raises the marketization level by 0.262 units, resulting in a 0.054 unit indirect improvement in environmental performance (0.262*0.207). The total effect (0.284) is the sum of direct and indirect effects, and the indirect effect accounts for 19%. Therefore, Hypothesis 1 is verified; that is, energy use rights trading policy can improve environmental performance by raising the level of marketization.

Similarly, the coefficients of energy use rights trading and technological innovation are significantly positive in the studies in columns (5) and (6), indicating that energy use rights trading can indirectly promote the improvement of environmental performance through technological innovation. Specifically, the indirect effect of energy use rights trading on environmental performance is 0.01, accounting for 3% of the total effect. Therefore, the energy use rights trading policy can improve environmental performance through technological innovation, and Hypothesis 2 is verified.

## 5. Conclusions and Policy Implications

### 5.1. Conclusions

In this study, 262 cities from 2005 to 2019 were selected as research samples and the difference model was used to conduct a comprehensive and detailed study on the impact of energy use rights trading policy on environmental performance. The main research conclusions are as follows: (1) The energy use rights trading policy can significantly improve the environmental performance of pilot cities. After a series of robustness tests, such as parallel trend test, propensity score matching, placebo test, triple difference method, and endogeneity test, the above results still hold. (2) The analysis of impact mechanism shows that the energy use rights trading policy can promote the improvement of urban environmental performance by promoting the improvement of marketization level and technological innovation ability. (3) Heterogeneity analysis shows that the impact of energy use rights trading policy on environmental performance will decrease with the increase in urban average income level; In resource-based cities, the impact on declining cities is the greatest, and the impact on growing cities is not significant. Simultaneously, the impact of energy use rights trading policy on improving environmental performance in old industrial bases is more visible than in non-old industrial bases.

### 5.2. Policy Recommendations

The government should continue to strengthen energy use rights trading policy and continuously improve the level of innovation driving. Environmental performance is a medium- and long-term development goal for the country, and the difference in green innovation intensity among enterprises can be taken into account in the allocation of energy use rights in different regions. Enterprises with high intensity of green innovation can appropriately increase the allocation of energy use rights, so as to stimulate more enterprises to invest resources to improve clean production technology and focus on the development of high-tech industries. It can not only promote the development of high-tech industry, but can also achieve the goals of reducing energy consumption and improving the environment. Simultaneously, the government should increase support for green innovation, especially for basic and forward-looking green innovation, including market support, policy support, and platform support.

In the aspect of building market conditions for the implementation of energy use rights trading policy, it should follow the market reform trend and give full play to the market attribute of the policy. To provide a good market trading platform, intermediary organizations, and legal support for energy use right trading subjects, particularly to address the synergistic effect between the government and the market in the implementation of energy use right trading policies. The government should not interfere in the implementation of the policy of trading energy rights and should strengthen the position of enterprises participating in trading energy rights as market players. The government can provide necessary regulation of energy use rights trading market, especially in paying attention to the cross-regional energy use rights trading policy design, and create a good business environment to provide sufficient factor market liquidity for energy use rights trading policy to improve environmental performance.

The government should provide the main body for trading energy use rights with a good market trading platform, an intermediary organization, and legal support, especially for dealing with the government and the market in trading synergy during the policy implementation process. In addition, the government may interfere with trade policy implementation, strengthen enterprise participation in trading market main body status, and provide the necessary energy-using trading market regulation. In particular, we will pay attention to the design of cross-regional energy rights trading policies, and create a sound business environment to provide sufficient factor market liquidity for energy rights trading policies to improve environmental performance.

We will pursue development in accordance with local conditions, maximizing the benefits of energy use rights trading policies. In the heterogeneity analysis of this study, the energy use rights trading policy have heterogeneous effects on the environmental performance of cities with different income levels, resource endowments and industrial types, and differentiated development policies should be formulated. On the basis of dynamic assessment of the changes in the types of resource-based cities, we should strengthen the implementation of energy use rights trading policy in declining and regenerative cities, strengthen the innovation incentive effect on the transformation of non-resource-based cities, and finally realize the maximum institutional dividend of energy use rights trading in improving environmental performance. In terms of urban reconstruction of old industrial bases, energy use rights trading policy should be applied to the reconstruction of heavy and chemical industries, high pollution, and high energy consumption old industrial bases. We will vigorously develop high-tech industries and modern service industries with low energy consumption and low pollution. It can not only improve the environmental performance of the old industrial base, but it can also provide technology spillover for the urban reconstruction of the old industrial base, achieving the goal of improving environmental performance.

### 5.3. Limitations

The study still has some limitations and should be expanded. First, from the aspect of research content, in order to conduct a more in-depth analysis of the energy use rights trading policy, more appropriate methods, and influencing mechanisms should be explored in future research. Second, in terms of research methodology, this study chose DID to examine the impact of the energy use rights trading policy on environmental performance. However, it lacked an analysis of the energy use rights trading policies spatial spillover effect. This should be further perfected in future research.

## Figures and Tables

**Figure 1 ijerph-20-03570-f001:**
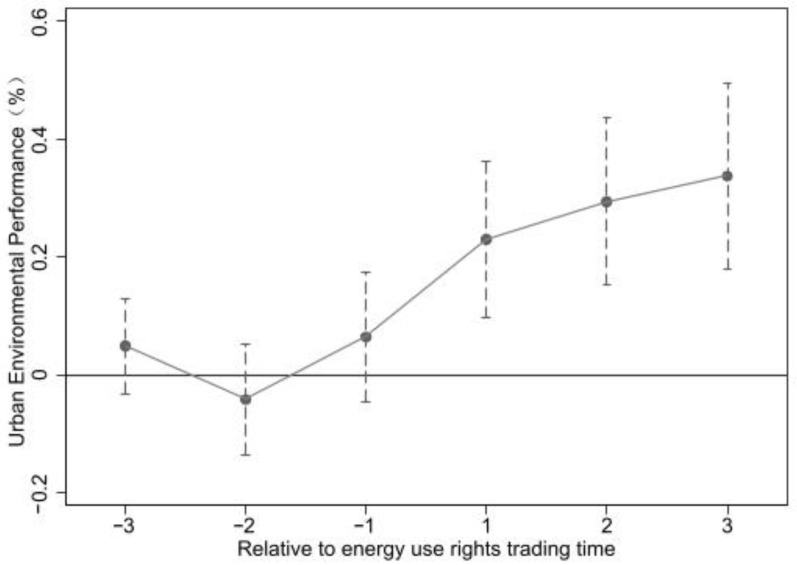
Parallel trend test.

**Figure 2 ijerph-20-03570-f002:**
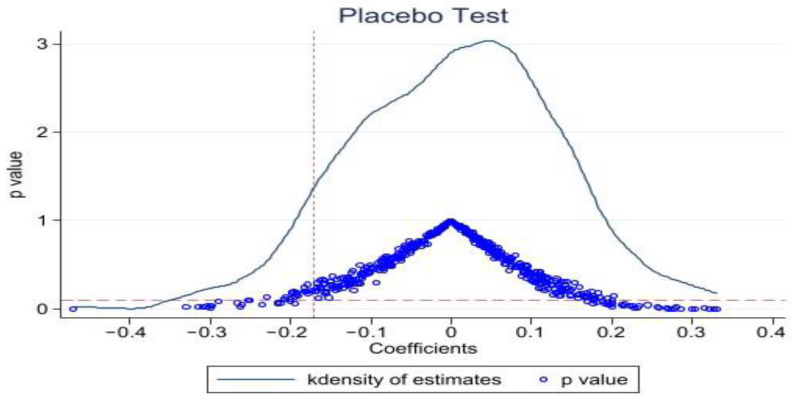
Placebo test.

**Table 1 ijerph-20-03570-t001:** Descriptive statistics of main variables.

Variable	Mean	Std.dev	Med	Min	Max
*lnep*	5.827	1.300	5.936	0.030	9.858
*ur*	0.344	0.230	0.283	0.039	1.026
*lfb*	180.275	194.288	130.546	4.932	2982.406
*structure*	0.395	0.662	0.377	0.086	41.390
*fin*	11.194	17.126	6.015	0.520	250.190
*market*	6.826	1.670	6.730	2.330	11.390
*innov*	4.499	0.692	4.690	1.120	5.300

**Table 2 ijerph-20-03570-t002:** Energy trading policy and environmental performance: Regression results of DID model.

	(1)	(2)	(3)
Variable	Regression of Environmental Performance Benchmarks		Dynamic Effects of Environmental Performance
*Energy use rights trading policy*	0.329 ***	0.284 ***	
	(0.042)	(0.041)	
*treat ∗ year2016*			0.417 ***
			(0.073)
*treat ∗ year2017*			0.287 ***
			(0.073)
*treat ∗ year2018*			0.225 ***
			(0.073)
*treat ∗ year2019*			0.205 ***
			(0.073)
*Control variables*	No	Yes	Yes
*Constant term*	6.002 ***	6.154 ***	6.292 ***
	(0.029)	(0.060)	(1.244)
*Time fixed effect*	Yes	Yes	Yes
*City fixed effect*	Yes	Yes	Yes
*R* ^2^	0.781	0.788	0.812

Notes: *** *p* < 0.01; Robustness standard errors in parentheses.

**Table 3 ijerph-20-03570-t003:** Heterogeneity analysis of different city sizes.

	Small- and Medium-Sized Cities	Large Cities	Super Large City
Variable	Environmental Performance		
*Energy use rights trading policy*	−0.014	0.319 ***	0.273 ***
	(0.362)	(0.058)	(0.054)
*Control variables*	Yes	Yes	Yes
*Time fixed effect*	Yes	Yes	Yes
*City fixed effect*	Yes	Yes	Yes
*observations*	112	2237	1311
*R* ^2^	0.755	0.790	0.818

Notes: *** *p* < 0.01; Robustness standard errors in parentheses.

**Table 4 ijerph-20-03570-t004:** Heterogeneity test of different resource-based cities.

	(1)	(2)	(3)	(4)	(5)	(6)
Variables	Environmental Performance					
	All resource-based cities	Growth city	Mature type city	Declining city	Regenerative city	Non-resource city
*Energy use rights trading policy*	0.617 ***	−0.043	0.355 ***	1.116 ***	0.989 ***	0.126 ***
	(0.070)	(0.273)	(0.089)	(0.181)	(0.201)	(0.049)
*Control variables*	Yes	Yes	Yes	Yes	Yes	Yes
*Constant term*	4.500 ***	7.081 ***	6.629 ***	4.250 ***	6.042 ***	6.907 ***
	(0.241)	(0.508)	(0.230)	(0.624)	(0.250)	(0.251)
*Time fixed effect*	Yes	Yes	Yes	Yes	Yes	Yes
*City fixed effect*	Yes	Yes	Yes	Yes	Yes	Yes
*observations*	1436	148	756	322	210	2224
*R* ^2^	0.729	0.736	0.729	0.762	0.736	0.822

Notes: *** *p* < 0.01; Robustness standard errors in parentheses.

**Table 5 ijerph-20-03570-t005:** Heterogeneity test between industrial bases and non-industrial bases.

Variables	Old Industrial Base	Non-Industrial Bas
	lnep	lnep
*Energy use rights trading policy*	0.550 ***	0.157 ***
	(0.076)	(0.047)
*Control variables*	Yes	Yes
*Constant term*	5.024 ***	5.566 ***
	(0.188)	(0.179)
*Time fixed effect*	Yes	Yes
*City fixed effect*	Yes	Yes
*observations*	1289	2371
*R* ^2^	0.722	0.848

Notes: *** *p* < 0.01; Robustness standard errors in parentheses.

**Table 6 ijerph-20-03570-t006:** Energy trading and environmental performance: estimation of instrumental variables.

Variable	First Stage Regression	Second Stage Regression
	Treat Period	lnep
*Iv period*	0.024 ***	
	(0.004)	
*treat period*		4.101 ***
		(0.704)
*Control variables*	Yes	Yes
*Constant term*	−0.050 *	3.925 ***
	(0.106)	(0.515)
*Time fixed effect*	Yes	Yes
*City fixed effect*	Yes	Yes
*observations*	3883	3883
*R* ^2^	0.374	0.157
*The F value*	17.510	

Notes: * *p* < 0.1, *** *p* < 0.01; Robustness standard errors in parentheses.

**Table 7 ijerph-20-03570-t007:** Robustness test of energy trading policy and environmental performance.

	(1)	(2)	(3)	(4)
Variable	Environmental Performance			
	PSM-DID	Triple difference method	Substitution of explanatory variables	Control for other policy implications
*Energy use rights trading policy*	0.310 ***		0.248 ***	0.246394 ***
	(0.040)		(0.049)	(0.041)
$period_{t}*treat_{t}*res_{t}$		0.366 ***		
		(0.049)		
*Control variables*	Yes	Yes	Yes	Yes
*Time fixed effect*	Yes	Yes	Yes	Yes
*City fixed effect*	Yes	Yes	Yes	Yes
*observations*	3598	3598	3598	2657
*R* ^2^	0.891	0.814	0.661	0.822

Notes: *** *p* < 0.01; Robustness standard errors in parentheses.

**Table 8 ijerph-20-03570-t008:** Impact mechanism test: Marketization level and technological innovation perspective table.

	(1)	(3)	(4)	(1)	(5)	(6)
Variable	lnep	lnep	market	Lnep	lnep	innov
*treat period*	0.284 ***	0.230 ***	0.262 ***	0.284 ***	0.278 ***	0.047 *
	(0.041)	(0.040)	(0.038)	(0.041)	(0.041)	(0.027)
*Market*		0.207 ***				
		(0.018)				
*Innov*					0.138 ***	
					(0.026)	
*Control variables*	6.154 ***	4.677 ***	7.139 ***	6.154 ***	5.535 ***	4.484 ***
	(0.060)	(0.144)	(0.056)	(0.060)	(0.130)	(0.040)
*Observations*	3883	3660	3661	3883	3659	3660
*R* ^2^	0.788	0.796	0.759	0.788	0.790	0.013

Notes: * *p* < 0.1, *** *p* < 0.01; Robustness standard errors in parentheses.

## Data Availability

The Ministry of Ecology and Environment of the People’s Republic of China (https://www.mee.gov.cn/, accessed on 15 August 2022); the National Bureau of Statistics (http://www.stats.gov.cn/, accessed on 16 August 2022); China National Intellectual Property Administration (https://www.cnipa.gov.cn/, accessed on 16 August 2022); and China Patent Information Center (http://www.cnpat.com.cn, accessed on 17 August 2022). EPSDATA (https://www.epsnet.com.cn/index.html#/, Home, accessed on 17 August 2022).

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
