# Peer review of "The Effect of Energy Use Rights Trading Policy on Environmental Performance: Evidence from Chinese 262 Cities"

_ijerph, 2023, doi:10.3390/ijerph20043570_

Round 1

Reviewer 1 Report

I think following comments will help to improve the quality of your paper.

The abstract of the study is a little awkward. A little modification of the abstract would help. For example, the last sentence could be a good fit to become the first sentence.

In the introduction section there is a paragraphs that needs citation(s). For example, the paragraph starts with "In 2016, the Chinese government". Please use bullet points or list to list your contributions.

In section 3, where did you get the benchmark model? Can you cite it?

Reviewer 2 Report

Review

The manuscript should be supplemented with more references. Write references in which the reader can become more familiar with the concepts used (for example, Coase property right theorem, Pareto optimal state of the energy, Porter's hypothesis, etc.). Write references for the methods used (Double difference method, Mediation effect test, etc.). Provide references for formulas in the paper.

It is necessary to write the manuscript more mathematically correctly. All variables have to be written in italic font. Some variables have indices, and some don't. It should be stated what variables mean in the formulas (for example, what does  represent in (1). Is the same as ?). By what index is the summing done in formula (4)? In what limits do other indexes change?

Correct spelling mistakes (especially in the abstract).

Write down the software used to obtain the results.

Improve the manuscript structure. The manuscript lacks a layout. Supplement the conclusion with directions for further research.

Reviewer 3 Report

According to the reviewer, the article is very interesting and raises a very important aspect related to reducing pollution of the second largest economy in the world. Any action to improve air quality in China will contribute to improving air quality around the world.

Round 2

Reviewer 2 Report

The manuscript can be accepted for publication.

Author Response

Thank you very much for your recognition of this study. The language expression of our full manuscript has been further improved and modified. Thank you again for your help and guidance.